# Emission Variations of Primary Air Pollutants from Highway Vehicles and Implications during the COVID-19 Pandemic in Beijing, China

**DOI:** 10.3390/ijerph18084019

**Published:** 2021-04-12

**Authors:** Xizi Cao, Ye Tian, Yan Shen, Tongran Wu, Renfei Li, Xinyu Liu, Amanzheli Yeerken, Yangyang Cui, Yifeng Xue, Aiping Lian

**Affiliations:** 1National Engineering Research Center of Urban Environmental Pollution Control, Beijing Municipal Research Institute of Environmental Protection, Beijing 100037, China; caoxizi@cee.cn (X.C.); shenyan980207@126.com (Y.S.); wutongran@cee.cn (T.W.); lrflrf930@163.com (R.L.); liuxinyu@cee.cn (X.L.); amanzheliyeerken@cee.cn (A.Y.); cuiyangyang@cee.cn (Y.C.); 2Beijing Municipal Ecology and Environment Bureau, Beijing 100048, China; yasutoko@163.com

**Keywords:** highway, coronavirus disease (COVID-19), emission variation, traffic flow

## Abstract

According to the traffic flow variation from January 2019 to August 2020, emissions of primary air pollutants from highway vehicles were calculated based on the emission factor method, which integrated the actual structure of on-road vehicles. The characteristics of on-highway traffic flow and pollution emissions were compared during various progression stages of coronavirus disease (COVID-19). The results showed that the average daily traffic volume decreased by 38.2% in 2020, with a decrease of 62% during the strict lockdown due to the impact of COVID-19. The daily emissions of primary atmospheric pollutants decreased by 29.2% in 2020 compared to the same period in 2019. As for the structure of on-highway vehicle types, the small and medium-sized passenger vehicles predominated, which accounted for 76.3% of traffic, while trucks and large passenger vehicles accounted for 19.7% and 4.0%, but contributed 58.4% and 33.9% of nitrogen oxide (NO_x_) emissions, respectively. According to the simulation results of the ADMS model, the average concentrations of NO_x_ were reduced by 12.0 µg/m^3^ compared with the same period in 2019. As for the implication for future pollution control, it is necessary to further optimize the structure of on-highway and the road traffic vehicle types and increase the proportions of new-energy vehicles and vehicles with high emission standards.

## 1. Introduction

The emission of oil-fired vehicles in transportation is an important source of urban atmospheric pollution [1,2]. As an essential carrier of transportation and part of the national road network, highways are significant means of regional passenger and freight transportation [3,4]. Under strong demand for passenger and freight transportation, there is a large on-highway traffic flow in Beijing, leading to considerable emissions of nitrogen oxide (NO_x_), hydrocarbons (HC), carbon monoxide (CO) and particulate matter with aerodynamic dynamiter less than 2.5 micrometer (PM_2.5_) from oil-fired vehicles, which have an impact on the surrounding atmosphere [5,6,7]. Therefore, identifying the characteristics of on-highway traffic flow and emissions and analyzing the causes of their variation under typical events are essential, which could provide a reference and support for atmospheric management and decision-making.

Quantitative characterization of on-road vehicle volume and emissions is a research hotspot [8]. For instance, Fan et al. constructed an oil-fired vehicles emission inventory of air pollutants in Beijing and analyzed the actual on-road pollutant emission characteristics based on the actual information of on-road traffic flow and emission factors [9]. Omid Ghaffarpasand et al. adopted the International Vehicle Emissions (IVE) model to estimate the vehicle emission factors in different regions of Isfahan, and identified the vehicle emission characteristics [10]. Davison et al. analyzed vehicle emission inventories and model contribution in the United States and Europe [11]. There has also been some research on the changes in pollutants of vehicles, influence factors, and control strategies [12,13], which showed historical changes in emissions and the effects of control measures. As the capital of China, Beijing has experienced rapid economic and social development and adopted a series of measures to control vehicle pollution [14,15]. The actual vehicle types and speed of on-road vehicles have undergone great changes. It is necessary to promptly identify the variation in the structure of the types and volume of on-road vehicles, further estimate the potential for emission reduction and put forward corresponding pollution control countermeasures.

There was an outbreak of a coronavirus disease (COVID-19) around the Spring Festival in 2020. In order to stop the spread of COVID-19, the government and society adopted the most stringent emergency control measures in history, which had a serious impact on human activities and the economy [16]. Therefore, it is very meaningful to study the effect of control measures on air quality, especially in transportation sector. According to the vehicle types and motor vehicle ownership, the emissions from vehicles on highways were measured based on the traffic flow data, which was obtained from the traffic monitoring stations from January 2019 to August 2020. By identifying the characteristics of the traffic flow and emission of vehicles on Beijing highways during COVID-19 pandemic, the environmental impact and emission reduction potential of vehicles on Beijing highways were analyzed. In addition, the emission reduction measures for vehicles on Beijing highways and implications and suggestions for future air pollution control are put forward.

## 2. Materials and Methods

### 2.1. Study Area

As the capital of China, Beijing is a large city with more than 21 million people living in an area of approximately 16,000 km^2^. The high population density and rapid economic growth have brought strong demand for travel and freight transportation. Even under many control measures in the transportation sector, the number of vehicles in Beijing still reached 6.1 million in 2018, and the annual passenger and freight turnover on highways reached 5.04 billion person-km and 16.74 billion ton-km, respectively [17]. Highways are both an important medium of transportation and a component of road networks [18]. Driven by the economic development in Beijing, the construction and development of highways are rapid. In 2018, the total length of highways reached 1114.6 km, accounting for 5% of the total mileage in Beijing [19]. The spatial distribution of the Beijing highways is shown in Figure 1, which includes 11 national highways and six other highways.

### 2.2. Data Sources

The data sources used in the study are presented in Table 1.

The information on hourly traffic flow, vehicle speed, monitoring mileage and vehicle types (small and medium-sized passenger vehicles, passenger vehicles, small trucks, medium trucks, large trucks, extra-large trucks and container trucks) was obtained from traffic monitoring stations on 17 highways from January 2019 to August 2020. The total length of the Beijing highways was collected from the Transport Development Annual Report. The meteorological information of surface temperature, wind speed, wind direction, precipitation rate and cloud cover was derived from the National Meteorological Information Center, China Meteorological Administration (http://data.cma.cn/, accessed on 15 January 2021).

The vehicle compositions of highways were obtained from road remote-sensing monitoring refer to the proportions of vehicles with different emission standards in China, which are shown in Figure 2. Motor vehicle emission standards are gradually tightening in China. The National I emission standard for vehicles has been implemented for nearly 20 years and the latest National VI was put into practice from 2020. At present, the average age of motor vehicles under National I to National VI emission standards is 19, 16, 12, 9, 4 and 1 years in order. It can be seen that all large trucks follow the National IV emission standard or above, while small/medium passenger vehicles, large passenger vehicles, small trucks and medium trucks buses whose vehicle fleet’s age are over 12 years (with emission standards of National III and below) still accounted for 4%, 6%, 13% and 11% of all vehicles, respectively, which have the most potential to reduce emissions and should be eliminated and updated in time.

### 2.3. Method of Calculating Emissions

Vehicle emissions were estimated using Equation (1) based on actual road traffic flow, road length and emission factors:(1)$$Q_{k}=\sum P_{k,i}\times M_{i}\times EF_{k,i}$$
where *k* and *i* represent the type of pollutant and vehicle, respectively; Q_k_ represents the emission; *P_k,i_* represents the number of type *i* vehicles and *M_i_* represents the annual mileage (km) of type *i* vehicles; and *EF_k,i_* represents the emission factor of pollutant *k* of type *i* vehicles (g/(km-vehicle)). The basic emission factors for different types of vehicles with different emission standards are all from Technical Manual for the Urban Air Pollutant Inventory [20]. Comprehensive emission factors weighted by emission standards and fuel percentage in Beijing were calculated and used in this study.

### 2.4. Simulation of the Impact of On-Highway Vehicle Emissions on Atmospheric Environmental Quality

The Atmospheric Dispersion Modeling System (ADMS-Urban) model can better deal with complex terrain and wind conditions of large cities based on atmospheric boundary layer parameters [21]. It considers the processes of gravity settlement, building topography and chemical reactions, then simulates multiple traffic pollution sources with different spatial distributions and finally obtains a higher-resolution pollutant concentration map to analyze the impact of different pollution sources on the prediction results [22]. It can accurately simulate not only the diffusion of pollutants emitted by various pollution sources in the atmosphere, including point, line and surface sources, but also their impact on the environment [23,24,25,26].

Compared with other atmospheric diffusion models, the weather processor in the ADMS model can automatically process various input data such as wind speed, date, time and cloud cover, or surface heat flux and boundary layer height to calculate boundary layer parameters. The obtained pollutant concentration prediction results are more accurate and credible. Therefore, the ADMS-Urban model was used in the study to simulate the influence of on-highway vehicle emissions on atmospheric environmental quality in Beijing between 2019 and 2020.

### 2.5. The Stages of COVID-19 Prevention and Control

Major public health emergencies of various degrees correspond to different response levels. According to the progression of the COVID-19 pandemic and people’s production and living, various levels of emergency responses were initiated in Beijing. January–August 2020 was divided into seven stages, which are depicted in Figure 3. The early stage of the pandemic was a period when COVID-19 had not yet spread and life was normal. Stage I of the level 1 response was the coexistence period of case input and local transmission, which was the most affected by the epidemic. Stage II of the level 1 response is the period of case input from other regions. The recovery stage of the level 2 response and stable stage of the level 3 response were the periods when anti-epidemic measures were normalized, and production and life were gradually restored. It was not until the second COVID-19 outbreak in June that Beijing gradually entered the stable stage of the downgraded level 3 response.

## 3. Results and Discussion

### 3.1. Characteristics of On-Highway Traffic Flow Variation during COVID-19 Pandemic

The data from traffic monitoring stations on 17 highways showed that the average daily traffic flow of Beijing in 2019 was 890,000. The average daily traffic flow in January–August of 2020 decreased by 38.2%. In particular, the average daily traffic flow dropped by 62% during the severe COVID-19 pandemic period (1.1–1.20). The outbreak and progression of the COVID-19 pandemic and the applied road traffic control measures were the main reasons for the decrease in traffic flow.

The on-highway traffic flow showed various trends during different outbreaks and progression stages of the COVID-19 pandemic (Figure 4). It can be seen that since the end of January, due to the impact of the COVID-19 pandemic, on-highway traffic flow decreased significantly. However, the traffic flow quickly rebounded and stabilized with the improvement of the local COVID-19 pandemic situation and the resumption of both social and economic activities.

In the early stage of the epidemic, the average daily traffic flow was 937,000, consistent with the same period in 2019. However, during stage I of the level 1 response, due to the strict control measures of COVID-19 pandemic, the average daily traffic flow decreased rapidly by approximately 61% as compared with the same period in 2019 and at the beginning of COVID-19 pandemic in 2020. The average speeds of on-highway passenger vehicles and trucks were 71 km/h and 56 km/h in 2020, respectively, which increased by 13% and 2%, respectively. Although Beijing delayed the restoration of regional traffic control measures during peak hours during COVID-19, there was no congestion on highways due to the decreasing traffic flow. In stage II of the level 1 response and the recovery stage of the level 2 response, traffic control returned to normal. The urgent and timely demand for materials led to a rapid increase by nearly 50% of the average daily traffic flow. In the stable stage of the level 3 response, the average daily traffic flow further restored due to the recovery of production and the normalized prevention measures of the COVID-19 pandemic, although it lasted a very short time in this stage. In mid-June 2020, a cluster outbreak of COVID-19 recurred in the Beijing Xinfadi Market Supermarket (stage of second COVID-19 outbreak); the traffic flow declined again by approximately 23% during the implementation of the secondary response, and the daily traffic flow recovered to 772,000 after stabilization. Vehicle control measures, such as flexible travel (any restriction to cross the city or types of vehicles can travel at certain period), played a major role in traffic flow reduction during the COVID-19 pandemic, greatly reducing the travel rate and allowing highways to operate smoothly.

The trends of the traffic volume on highways in Beijing between 2019 and 2020 at each highway monitoring station during the prevention and control of COVID-19 pandemic are shown in Figure 5. As a transit corridor between Beijing and Chengde and a major highway in suburban areas, the Jingcheng Highway still had the largest cumulative traffic flow (9.8%), which was the least affected by the COVID-19 pandemic, undergoing an 8.3% reduction in traffic flow. The inter-annual variability in the structure of the on-highway vehicle types was small; small passenger vehicles predominated. Passenger vehicles and trucks accounted for 79.8% and 20.2% of the traffic, respectively.

### 3.2. The Variation and Spatial Distribution Characteristics of Atmospheric Pollutants on Highways during the COVID-19 Pandemic

According to the emission intensity of vehicles, the four primary and typical types of air pollutants (CO, HC, PM_2.5_, NO_x_) with larger emissions were obtained combined with the current structure of vehicle types in Beijing and the calculation method in Section 2.3. The emission of one single pollutant, NO_x_, during different stages of the COVID-19 pandemic was presented in Figure 6. During the strict prevention and control period of the COVID-19 pandemic, the pollution emissions decreased significantly, by nearly 1/3rd compared with the same period in 2019, and reached their minimal value in the stages of the level 1 response, with a reduction of 57.4%. However, with the improvement of the local COVID-19 pandemic situation and the resumption of both social and economic activities, the pollution emissions on highways rapidly rebounded.

Based on the geographical location characteristics of various types of pollution sources, the spatial allocation emission on Beijing highways were used to produce a 1 km × 1 km grid of pollutants for the 2019, which is shown in Figure 7. It can be indicated that the four pollutants were mainly concentrated in the northeast Beijing. Among them, the pollutant emission intensity on the Jingcheng (JC) Highway was the highest. The emission intensity of the four pollutants (NO_x_, PM_2.5_, CO, HC) in 2020 were 80–100 kg/(d·km), 2.4–2.6 kg/(d·km), 140–200 kg/(d·km) and 32–35 kg/(d·km), respectively, which had a decrease of 23–29% compared to 2019. It is related to the large accumulative traffic flow of passenger vehicles and truck. On the contrary, the pollutant emission intensity on the Jingkun (JK) Highway was the lowest, with an emission density of 1–40 kg/(d·km), 0.1–0.8 kg/(d·km), 1–60 kg/(d·km) and 1–8 kg/(d·km), respectively, showing a decrease of 32–38% compared to 2019. Compared with 2019, the emission intensity of all highways showed a significant downward trend during the COVID-19 pandemic, and they declined at different levels. Among them, the Capital Airport (CA) Highway was significantly affected by the COVID-19 pandemic; the emission intensity of the four pollutants has decreased by 47–52%, caused by the travel reduction of people and cancellation of flights.

According to the data on the variations of ambient air pollutants concentration nearby the highway monitoring sites, the air pollutants (CO, NO_2_, PM_2.5_) on each highway in 2020 showed various degrees of decline compared to 2019, with a decrease of 8–18%, of which NO_2_ is the most obvious. It can be seen that the reduction of vehicle emissions contributed to the improvement of air quality. From the perspective of spatial distribution, strengthening highway-related emission regulations by restricting vehicle types, improving emission standards on highways and time-splitting traffic control measures can optimize the structure of on-highway vehicle types and decrease the vehicle flow, thereby reducing emissions and mitigating the overall environmental impact. Future on-highway vehicle emission regulation depends on highways with relatively high truck traffic flow, such as the Jingcheng Highway, Jingzang Highway and Jingxin Highway.

As is shown in Figure 8, the contributions of different types of vehicles to pollutant emissions can be compared. The small and medium-sized passenger vehicles predominated in the on-highway traffic flow and were the major contributors to CO and HC, accounting for 51.7% and 71.5% of total emissions, respectively. The proportions of large passenger vehicles and large/medium trucks were small. Due to their high emission intensity per vehicle, large passenger vehicles contributed 15% of HC and 27% of CO despite accounting for only 4% of traffic flow. Medium and large trucks, accounting for 13% of traffic flow, contributed nearly 53% of NO_x_ and 37% of PM_2.5_. Large passenger vehicles and large/medium trucks are important vehicles to target for pollution reduction control on highways. The pollutant emission can be improved by controlling the structure of on-highway vehicle types through such policies as restricting traffic, banning high-emitting vehicles, encouraging low-emitting vehicles and developing rail transportation, electric transportation and clean transportation.

### 3.3. Discussion of Air Quality

The daily average pollutant emissions on highways were regarded as the input sources, combining the hourly meteorological data from the Nanyuan meteorological station of 2019. The ADMS-Urban model was used to simulate the impact of the pollution emissions on Beijing highways on surrounding atmospheric quality from January to August between 2019 and 2020 (Figure 9). It could be seen that the emission concentrations of PM_2.5_ and NO_x_ on 17 highways in Beijing decreased, the influencing range of pollutant diffusion narrowed and the area with high concentrations of relevant pollutants shrank in 2020, which were all caused by the prevention and control requirements of the COVID-19 pandemic and related pollutant control measures. Moreover, according to the result of the simulation, the most affected point by the highway is located at the junction of the Shunyi, Huairou and Changping districts, which is 25 m away from the Jingcheng Highway. The densely polluted areas on highways are heavily populated, and serious air quality problems will greatly affect human health to some extent. Compared to 2019, the concentrations of NO_x_ and PM_2.5_ were 59.7 μg/m^3^ and 1.5 μg/m^3^, respectively, with a decrease of 20.9% and 21.1%.

Compared with the striped distribution of pollutants on highways, the spatial distribution of pollutants in the population center is more concentrated, mainly in the areas within the Fifth Ring Road of Beijing. Areas with a high concentration of pollutants in the downtown area are especially notable. The spatial distribution of pollutants in Beijing reveals the trend of outward radiation from central urban area. The influence range of pollutants on highways entering and leaving Beijing is significantly larger than that of other areas. Meanwhile, the large passenger and freight vehicles passing through the highway at night become the main source of pollution in Beijing. Moreover, the distribution of the pollutants changes to the tendency based on the transfer from the outside into the urban areas.

### 3.4. Analysis of Emission Reduction Potential and Implications for Future Pollution Control

The analysis showed that the emission reduction potential of on-highway vehicles was concentrated in small passenger vehicles, which had a large traffic flow, and large passenger vehicles or medium/large trucks, which had a high emission intensity per vehicle. The specific implications for pollution control are as follows:

(1) Adjust the structure of passenger vehicle types. The use intensity of passenger vehicles on highways was high, accounting for approximately 80% of the traffic flow and contributing approximately 87% of the HC and 78% of the CO emitted. Therefore, passenger vehicles have the huge potential for HC and CO emission reduction. The emission standards for in-use vehicles can be strictly enforced through a combination of encouragement and enforcement measures. Corresponding restrictions and emission supervision can be imposed on overaged vehicles, since old vehicles with National III and below emission standards in 2020 still contributed to 31.5%, 24.0% and 12.8% of the NO_x_, PM and HC emissions, respectively.

(2) Optimize the structure and improve the emission standards of trucks. Trucks accounted for approximately 20% of on-highway traffic flow, and diesel trucks contributed to 50–60% of the emission of NO_X_. Since old diesel trucks with National III and below emission standards in 2020 still contributed to 12.9%, 9.7% and 2.2% of the NO_x_, PM and HC truck emissions, respectively, the replacement and elimination of these can be promoted through different policies to promote the use of energy-saving and environmentally friendly or new-energy trucks. For long-distance freight transportation, the existing railway transportation resources should be fully used and the proportion of railway freight transportation are supposed to be increased. Moreover, it is necessary to establish an urban green freight transportation system.

(3) Improve the supporting facilities and strengthen highway-related emission regulations. The cumulative traffic flow on highways in Beijing reaches 300 million per year. Highways with large traffic flow, such as the Jingcheng Highway, Jingzang Highway and Jingping Highway, often suffer from serious congestion. Therefore, traffic-supporting facilities should be improved. For example, the non-parking-toll system, unattended detection system for traffic violations on highways and vehicle-mounted dynamic monitoring system should be promoted. Future on-highway vehicle emission regulation can be strengthened by restricting vehicle types, improving emission standards and time-splitting traffic control measures.

(4) Strengthen supervision and law enforcement during key periods. The spatial distribution of highway traffic flow exhibits different patterns. For the Jingkai Highway, Jingha Highway and Jinghu Highway in the southeastern region of Beijing, the traffic flow of high-speed trucks is relatively high, so that dynamic monitoring devices can be set up at key nodes of the road network, such as sections with large traffic flow, monitoring stations for traffic violations and district/county entrances, in order to improve the monitoring network. At the same time, in order to reduce pollution, vehicles that do not meet emission standards ought to be eliminated and updated promptly. Moreover, daily maintenance is supposed to be strengthened to ensure that the emissions reach the standards during the actual usage stage.

## 4. Conclusions

Based on the rapid variation in the structure of vehicle types and on-highway traffic flow in Beijing from January 2019 to August 2020, the vehicle emissions and their causes, by comparing different outbreaks and progression stages of the COVID-19 pandemic, were analyzed. By determining the actual structure of on-road vehicle traffic and its influencing factors, targeted emission reduction vehicles were analyzed. Moreover, the implications for future pollution control were put forward. The main conclusions are as follows:

(1) The short-term strict control measures during the COVID-19 pandemic drastically reduced both traffic flow and emissions. Compared with passenger vehicles flow with 14.1% reduction, trucks were more affected, showing a reduction of approximately 28.7%, and the emission intensity of four pollutants was reduced by approximately 30%, amounting to a good emission reduction effect. However, with the improvement of the local COVID-19 pandemic situation and the resumption of both social and economic activities, the traffic flow and emissions on highways quickly recovered.

(2) The COVID-19 pandemic had little effect on the structure of on-highway traffic, as the traffic flow of medium-sized and small passenger vehicles always accounted for the highest proportion. In terms of the emission contributions, large passenger vehicles and trucks emitted relatively large amounts per vehicle. Large passenger vehicles, accounting for only 4% of the traffic flow, contributed 15% of HC and 27% of CO, while trucks, accounting for 13% of the traffic flow, contributed nearly 59% of NO_x_ and 43% of PM_2.5_.

(3) In the spatial distribution, the Jingcheng highway was the highest contributor to the emission intensity due to a large accumulative volume of passenger vehicles and trucks. In the longitudinal comparison, the Capital Airport Highway was most affected by the COVID-19 pandemic due to flight cancellations, and decreases in traffic flow and emission intensity were the most obvious.

(4) To reduce air pollution from on-highway vehicles, on the one hand, the emission standards of vehicles should be strictly enforced, and the structural adjustment and optimization of diesel trucks are supposed to be promoted to improve emissions. On the other hand, in order to reduce vehicle emissions and improve traffic congestion, supporting facilities and law enforcement during key periods ought to be enhanced, and dynamic monitoring systems at key nodes can be set up to strengthen supervision.

## Figures and Tables

**Figure 1 ijerph-18-04019-f001:**
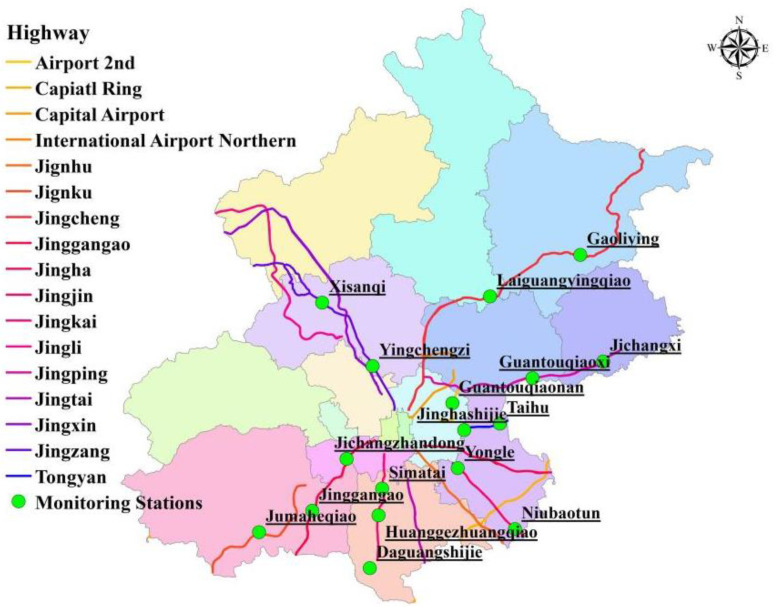
Spatial distribution of highways in Beijing.

**Figure 2 ijerph-18-04019-f002:**
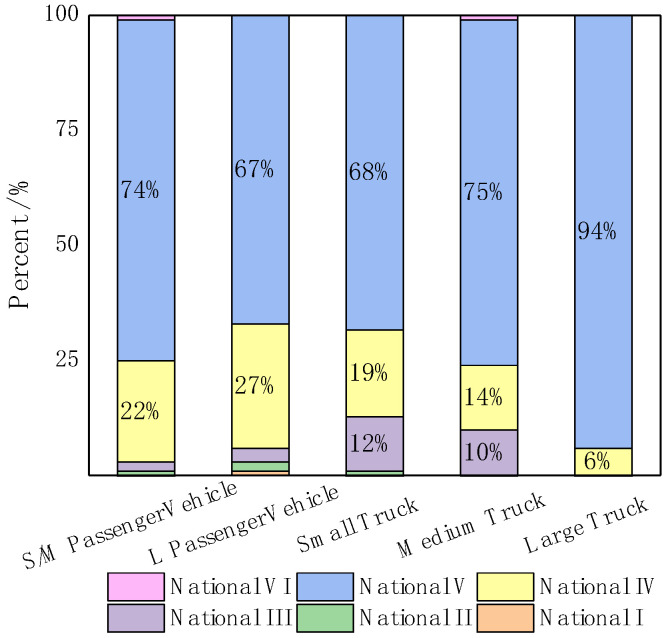
Composition of the emission standards of each type of vehicle in Beijing.

**Figure 3 ijerph-18-04019-f003:**
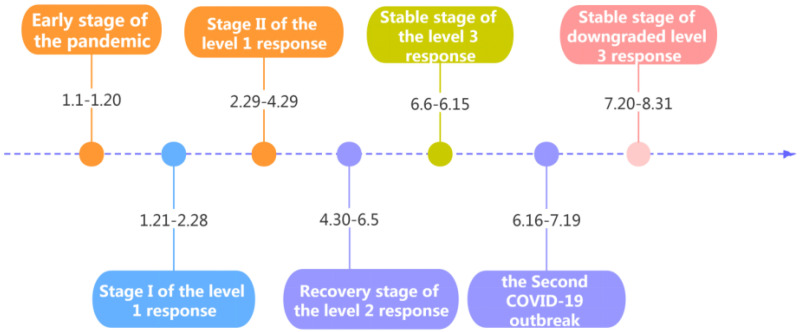
Classification of the different stages of the COVID-19 pandemic.

**Figure 4 ijerph-18-04019-f004:**
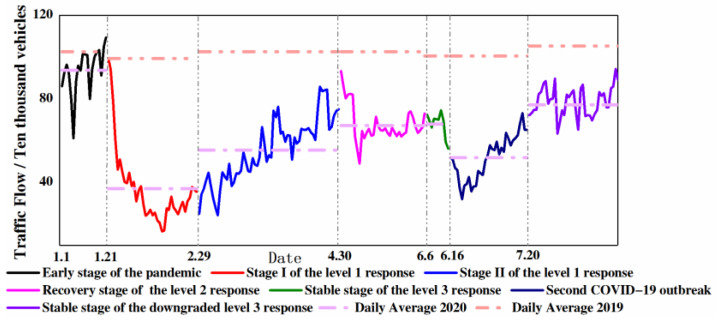
Trend of traffic flow on the highways in Beijing during different stages of the COVID-19 pandemic.

**Figure 5 ijerph-18-04019-f005:**
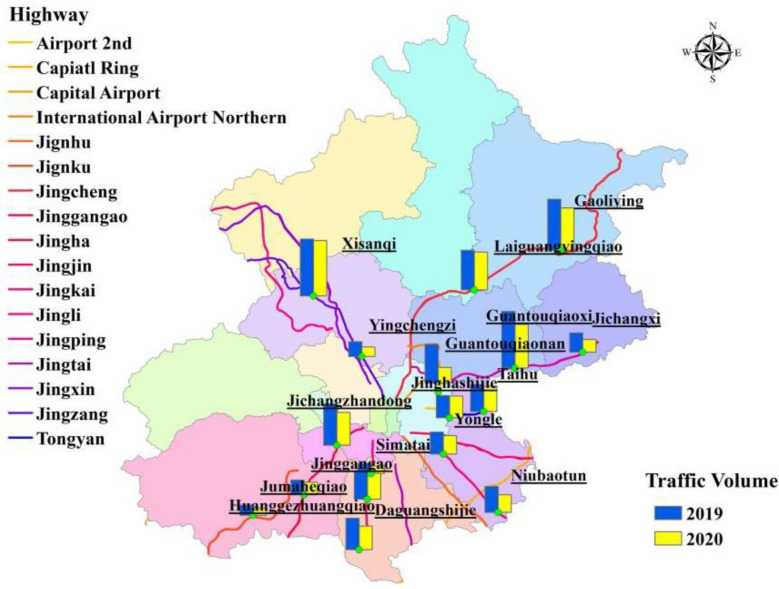
Trend of traffic volume on highways in Beijing between 2019 and 2020.

**Figure 6 ijerph-18-04019-f006:**
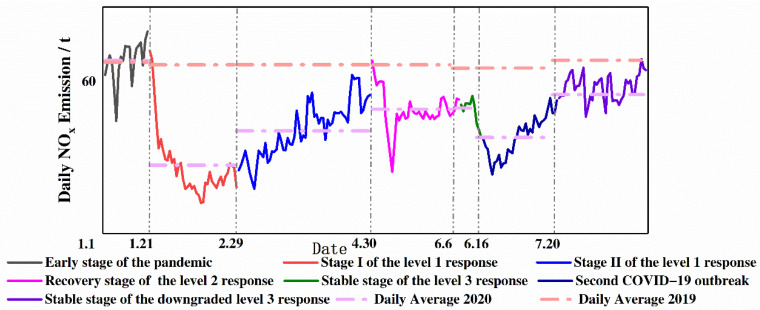
Daily NO_x_ vehicle emissions on highways in Beijing during different stages of COVID-19 pandemic.

**Figure 7 ijerph-18-04019-f007:**
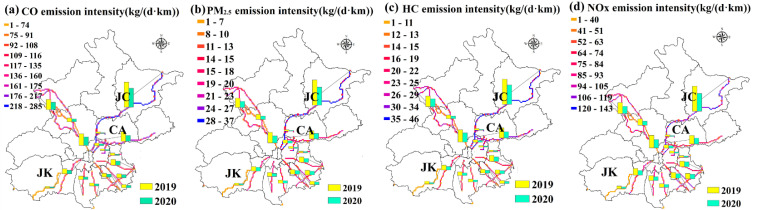
The spatial distribution of pollutant emission intensity on highways between 2019 and 2020 in Beijing.

**Figure 8 ijerph-18-04019-f008:**
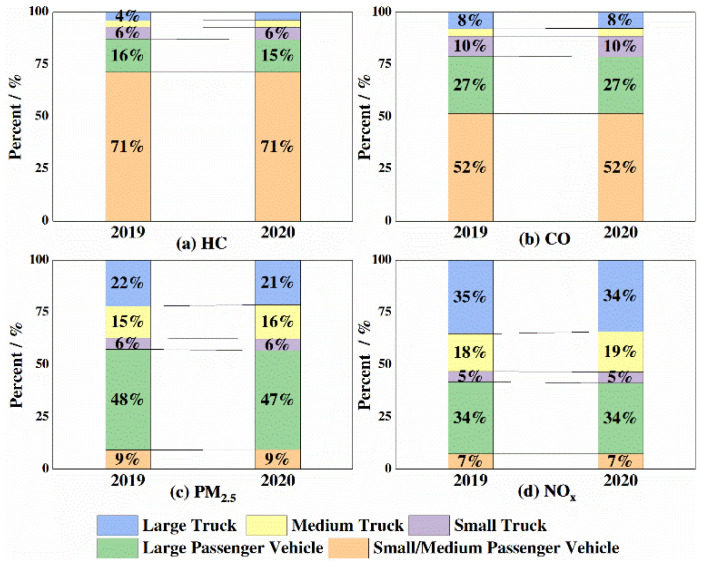
Contribution of different types of vehicles to pollutant emissions.

**Figure 9 ijerph-18-04019-f009:**
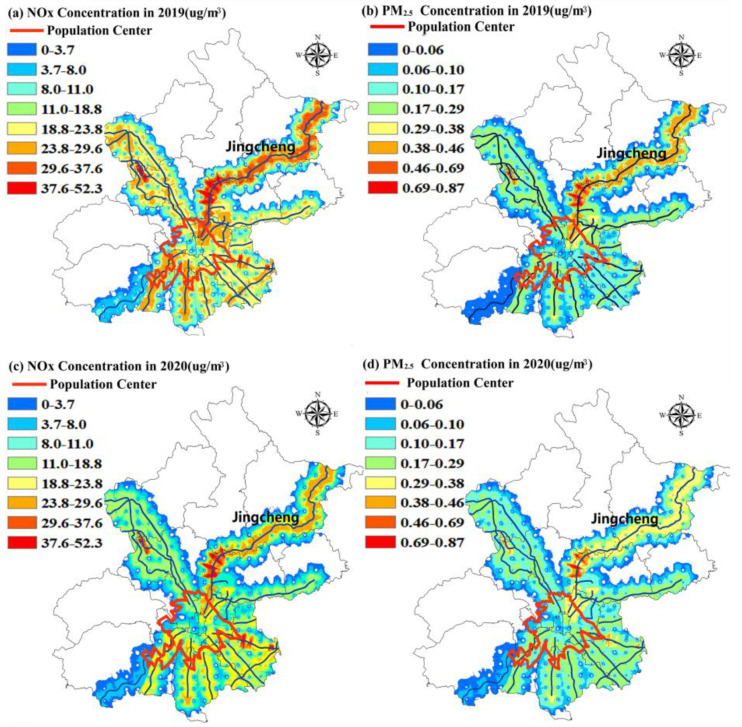
Effect of pollutant emission on Beijing highways on air quality.

**Table 1 ijerph-18-04019-t001:** Types of data and sources of emission from databases in Beijing.

Data Types	Sources
Traffic flow, speed, vehicle types	Traffic monitoring stations on highways
The structure of on-highway vehicle types (the proportions of vehicles with different emission standards)	Road remote-sensing monitoring
The length of highways in Beijing	2019 Transport Development Annual Report (Beijing Transport Institute, 2019)
The meteorological information	National Meteorological Information Center, China Meteorological Administration

## Data Availability

Not applicable.

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
