# Peer review of "Emission Variations of Primary Air Pollutants from Highway Vehicles and Implications during the COVID-19 Pandemic in Beijing, China"

_ijerph, 2021, doi:10.3390/ijerph18084019_

Round 1

Reviewer 1 Report

This paper presents the impact of COVID-19 on the monitored traffic flows on the highways of Beijing as well as modelled changes in air pollutant emissions. The topic is interesting, but the monitoring data and this study design don’t give basis for too much conclusions. I think some restructuring of the study design is needed to add to the value of the monitoring data.

General comments:

  • The paper is written in English that is understandable and for the most part easy to read, but still has grammatical errors that need to be fixed by a native speaker. Too often a key point is left somewhat ambiguous due to this.
  • In the current study design the traffic flow, emissions estimates and recommendations are somewhat detached from each other. This is the main issue of the manuscript. Fig 4 (traffic flow) and Fig 6 (emissions) have basically identical curves, meaning that the structure of the vehicle fleet didn’t really change due to COVID-19 (this was also written in the paper). This does not give much tools for further analysis, although it’s an interesting discovery. Since the changes in the emissions have not been validated in any way with measurements, nor have they been used to estimate population exposure resulting from changes in concentrations, I don’t see the emission estimates bringing much of an extra value here. Although I don’t disagree with the recommended mitigation measures, I don’t think these measures can be suggested as a conclusion of this specific study.

Specific comments by line numbers

#31 Title “Introduction” missing

#84 The font in the legend is a bit difficult to read. Same goes for most figures in the manuscript.

#100 These national emission standards could be briefly explained for non-Chinese audience. E.g. what do IV and V mean in terms of vehicle ages?

#136 The different stages and response levels of the pandemic should be explained.

#146 Unit should be added to the average traffic flow.

#192 What pollutants is the Figure presenting?

#201 Why were these specific pollutants chosen for the analysis?

#217 How would a highway-specific emission regulation work and would it mitigate the overall environmental impact of traffic emissions?

#233 What are the implications of changing air quality to population exposure? Even a qualitative analysis would add value (e.g. what is the spatial distribution of population centers in relation to these highways?)

#248 One thing that could be analyzed with the available data is the effect of the vehicle fleet’s average age on emissions. On the other hand, I don’t see the connection between some of these recommendations and the results of this study.

Reviewer 2 Report

The main scope of the article is to present the effects on road emissions due to the COVID-19 pandemic. An interesting approach was performed by using traffic flow data as well as the vehicles type’s characteristics. The calculated emissions were further used as input at the ADMS-Urban model in order to study the impact on the local air quality.

Generally the manuscript is well arranged in four sections (a title should be added for the introduction section) and the proposals made at the end of the manuscript towards the road emissions reductions are of great interest. However, corrections are either needed or required and clarifications should be provided in order to improve the paper to be accepted. Please read the following comments.

Point 1: Highlights should be rewritten providing the most important outcomes of the study and not information about the methodology.

Point 2: The fonts at most of the figures should increase.

Point 3: lines 60 – 64. Please rephrase. The sentence is too complex.

Point 4: The introduction should be rewritten and enriched with more references (from the international scientific community) concerning the road emissions inventories.

Point 5: line 87. Please correct the sentence "The data sources used in the study are as follows” as: "The data sources used in the study are presented in table 1".

Point 6: Table 1. Please mention to which data the word "type" refers to. Types of vehicles?

Point 7: Since vehicles speed data are available a separate section or paragraph could be added in the manuscript with comments on the relevant changes for 2019 to 2020.

Point 8: lines 95 – 97. Please mention the temporal scale the meteorological parameters were provided. How many meteorological stations were used in order to provide the meteorological parameters?

Point 9: lines 98- 100. Please add more comments for the composition of the vehicles engine technology presented in figure 2. Are data concerning the fuel type used by each vehicle category available? It would be useful to include such information in the text since different fuel types are related with certain pollutants.

Point 10: lines 112 – 113. Are these emission factors fuel specific also? Please comment.

Point 11: Section 1.5. It would be useful to add more comments for the separation of the COVID-19 pandemic period into the 3 levels and relate them with the restrictions in people's and vehicles movements. For instance, how the lock-down measurements affected the different vehicles types? I suppose that the majority of passenger cars were out of use. Please comment.

Point 12: Figure 5. Please increase the fonts. It is not clear whether the bars refer to emissions or traffic volume. In figure 5 it is mentioned as emissions but in the legend and the text as traffic volume. Please clarify.

Point 13. Section 2.2. Please provide the methodology followed for the spatial allocation of emission on highways in order to be used as input for the air quality application.

Point 14: Figure 7. Please increase the fonts. The colour scale is the same for the figures representing the same pollutant? To which period these figures refer? Please clarify if the emission values are mead daily, for weekday, weekend?

Point 15: lines 205 – 218. Please add the difference in absolute value or percentage. In general, please add values to support the mentioned differences from 2019 to 2020, from region to region.

Point 16: lines 218 – 230. Which is the fuel type that all these vehicles categories use? Is it the same? If not, the fuel might also affect the contribution to different pollutants.

Point 17: Section 2.3. Please mention the simulation period. e.g. From December 2019 to June 2020?

Round 2

Reviewer 1 Report

The revision has notably increased the clarity and value of this paper. However, the English language was not edited. This needs to be done before publication. In addition, to further improve readability and interest to the readers, the following specific comments should be addressed.

Fig 1: Even now that the texts are visible, the connection between the colors, legend and texts in the map is still quite unclear. The same goes for Fig 5. They need restructuring. I also don’t understand the vertical blue and yellow bars in Fig 5 below the text “traffic volume.” To me it indicates that the volumes increased notably in 2020

#201: “As a transit corridor between Beijing and Chengde and a major high-201 way in suburban areas, the Jingcheng Highway still had the largest cumulative traffic flow 202 (9.8%), which was the least affected by the COVID-19 pandemic, undergoing a 22% re-203 duction in traffic flow.”

If I’m looking at the right bars, I see a much smaller reduction than 22%?

#208: Can it be concluded that a 1% increase is related to anything specific?

Fig 6: Does the graph represent the sum of all those emission components? I think selecting one (like PM or NOx) would be more appropriate

Fig 7: It’s very difficult to compare the colors between 2019 and 2020 and as such the second row of figures doesn’t give much information. Maybe it could be replaced with maps of percentual decrease of emissions from 2019. In addition, the colors should be more distinctive overall

#247: Can the improvement of emission standards be called highway-specific regulation?

Fig 9: Legends are too small. Without knowledge of the city, it’s difficult to place the mentioned population centers on the map. They should be shown in the Fig.

#304: “Corresponding restrictions and emission supervi-304 sion can be imposed on overaged vehicles and vehicles with low emission standards.”

How much did these old vehicles contribute to the overall emissions?

#308: “Since old diesel trucks still contribute a great number of emissions”

How much?

#338: “the emission reduction potential was identified”

To me this implies that the amount of potential reductions (e.g. in tons) for some specific measures was identified

#341: “Truck traffic flow was affected significantly”

Wasn’t all traffic affected pretty much the same?

Reviewer 2 Report

The authors revised the manuscript following reviewers' comments. The necessary corrections were made and satisfactory explanations were given. As a consequence the enriched text meets requirements for publication after making some minor corrections.

  • Fonts in figure 9 should increase
  • Highlights should be written in order to provide the main outcomes of the research. Especially the highlight "Reduced traffic and emissions improved surrounding atmosphere of highways during the COVID-19"should be rephrased.
  • There are some minor grammatical errors in the text
